# Biaxial Piezoelectric MEMS Mirrors with Low Absorption Coating for 1550 nm Long-Range LIDAR

**DOI:** 10.3390/mi14051019

**Published:** 2023-05-09

**Authors:** L. Mollard, J. Riu, S. Royo, C. Dieppedale, A. Hamelin, A. Koumela, T. Verdot, L. Frey, G. Le Rhun, G. Castellan, C. Licitra

**Affiliations:** 1University Grenoble Alpes, CEA, Leti, F-38000 Grenoble, France; 2Beamagine, Bellesguard, E08755 Castellbisbal, Spain; 3Centre for the Development of Sensors, Instrumentation, and Systems, Universitat Politècnica de Catalunya (UPC-CD6) Rambla Sant Nebridi 10, E08222 Terrassa, Spain

**Keywords:** quasi-static actuator, 2D MEMS mirror, piezoelectric, Bragg reflector, high power management

## Abstract

This paper presents the fabrication and characterization of a biaxial MEMS (MicroElectroMechanical System) scanner based on PZT (Lead Zirconate Titanate) which incorporates a low-absorption dielectric multilayer coating, i.e., a Bragg reflector. These 2 mm square MEMS mirrors, developed on 8-inch silicon wafers using VLSI (Very Large Scale Integration) technology are intended for long-range (>100 m) LIDAR (LIght Detection And Ranging) applications using a 2 W (average power) pulsed laser at 1550 nm. For this laser power, the use of a standard metal reflector leads to damaging overheating. To solve this problem, we have developed and optimised a physical sputtering (PVD) Bragg reflector deposition process compatible with our sol-gel piezoelectric motor. Experimental absorption measurements, performed at 1550 nm and show up to 24 times lower incident power absorption than the best metallic reflective coating (Au). Furthermore, we validated that the characteristics of the PZT, as well as the performance of the Bragg mirrors in terms of optical scanning angles, were identical to those of the Au reflector. These results open up the possibility of increasing the laser power beyond 2W for LIDAR applications or other applications requiring high optical power. Finally, a packaged 2D scanner was integrated into a LIDAR system and three-dimensional point cloud images were obtained, demonstrating the scanning stability and operability of these 2D MEMS mirrors.

## 1. Introduction

For automotive, biomedical or entertainment applications, such as pico-projection, MEMS mirrors are of great interest due to their small size, low cost and low power consumption compared to standard mechanical beam scanning systems. Many publications present the potential application of MEMS scanners and their requirements [1,2]. Current works focus on the development of long-range LIDAR systems which are considered essential sensors for autonomous driving, as they can provide accurate three-dimensional information in the form of high-density point clouds. Indeed, based on the same principles as radars, but using shorter wavelengths, LIDAR systems offer better spatial resolution. For long-range applications (a distance greater than 100 m), the MEMS-LIDAR approach, based on a MEMS-mirror, currently seems to be the best option. In comparison, scanning based on optical phased arrays (OPA) or Flash-LIDAR systems suffer from limitations relating to the required beam power [3,4].

Since long-range detection requires high laser power, eye safety considerations are of primary importance. Eye safety depends on many parameters, such as wavelength, laser power and beam divergence. Although a LIDAR system operating at a wavelength of 905 nm has advantages in terms of the cost and availability of lasers/detectors. At this wavelength, the eye transmits the incident light to the retina with negligible attenuation, limiting acceptable beam power and making long-range detection difficult to achieve. Operating at a wavelength of 1550 nm is, therefore, an attractive option as it can allow the use of a 10-fold increase in power, compared to a 905 nm system without excessing incurring eye safety problems [4].

To drive the MEMS mirror, four main actuation mechanisms are common: thermal, magnetic, electrostatic and piezoelectric. Most commercial MEMS scanners use either electrostatic or electromagnetic actuators [1,2]. Each of these actuation mechanisms have both advantages and drawbacks. Thermal actuation can achieve large deflection using low driving voltages, but its power consumption is high compared to other driving mechanisms and actuation frequencies are limited by the thermal response time. Electromagnetic actuation can provide high force and also requires low driving voltage but requires costly and bulky external magnets. Additionally, electromagnetic interference increases the overall component cost as specific mirror protections is required to maintain functionality. Electrostatic scanners can be fabricated using well-established manufacturing technologies and provide attractive mechanical performance. However, high driving voltages are required, typically greater than 150 V, and the comb-finger capacitors can cause high lateral damping in air, requiring vacuum packaging to improve mechanical efficiency. Piezoelectric actuators often suffer from a comparatively small range of movement which may limit the range of optical angles that can be achieved for a beam scanner. The piezoelectric technology has the advantage of providing the possibility of a monolithic beam scanner without the need for further fabrication steps, such as separate integration of magnets integration. This technology also allows low voltage actuation (<25 V) for quasi-static (non-resonant) operation. In addition to other advantages, such as fast response times and low power consumption, piezoelectric mirrors appear as attractive candidates for long-range LIDAR applications.

A key challenge was to design and fabricate the mirror in such a way that it can withstand the required incident optical laser power. The system specifications dictate an average optical power of 2W over the 2 × 2 mm^2^ side mirror. The laser has a pulse duration of 3 ns, and a pulse repetition rate (PRR) of 120 kHz which represents an instantaneous peak power of nearly 5.5 kW. The reflective layers are usually metallic (mostly aluminium or gold) regardless of the type of actuation of the MEMS scanner: piezoelectric [5,6,7], electromagnetic [8] or electrostatic [4]. For non-polarized incident light with a wavelength of 1550 nm, the reflectivity of gold is about 99% that of aluminium is near 97%. This means that the power absorbed by the metal layer corresponds to about 1% of the incident power in the case of gold and 3% in the case of aluminium. Because of the heating of the mirror, this absorption limits the maximum possible incident optical power. Notwithstanding the possibility of irreversible mirror damage, mirror overheating leads to thermal deformation of the mirror which affects the form of the reflected beam as well as hinders the performance of the piezoelectric material. Furthermore, in the case of a technological process using a bonded mirror, the compatibility of the bonding with potential overheating must be taken into account.

For these reasons, in order to achieve higher beam power, we have developed a multi-layered non-metallic Bragg reflector that limits incident power absorption and is fabricated using a monolithic silicon fabrication process. The development of a Bragg reflector on a 2D MEMS scanner has already been published but with electrostatic actuation technology and for a different wavelength [9] or by transferring a Bragg mirror on a piezo substrate [10]. The former is a different technological process from using sol-gel piezoelectric material and the latter is not a monolithic and collective approach.

To the best of our knowledge, this paper will show for the first time the fabrication, using VLSI technology, of a 2D piezo-scanner designed for a 1550 nm LIDAR system, using a low-absorption dielectric reflective coating compatible with sol-gel deposition of the piezoelectric motor. In the following, we will compare 2D MEMS piezo-scanner with a Bragg and a gold reflector to ensure that the piezoelectric material performances are preserved.

## 2. Design and Fabrication

### 2.1. Scanner Specifications

We developed the MEMS scanner to be integrated into an existing LIDAR system, it, therefore, had to be designed with a given package size. Because of this fixed volume constraint, a biaxial scanner was chosen which is both compact and easier to package than a multi-mirrors approach. The second key specification was the MEMS mirror size, which defines the angular resolution of the LIDAR system. The angular resolution Δθ is the smallest angular distance between two resolved points. It is defined as $\Delta \theta =K\frac{\lambda }{D}$ where λ is the operating wavelength and *D* is the width/height of the square-shaped mirror. *K* is an aperture factor that will be taken as equal to one in this case [11]. The requirement of the LIDAR system was, in our case, to discriminate an object of a height of 20 cm (such as a tire) at a distance of 100 m. The beam angular resolution must, therefore, be less than 0.05° which requires a 2 mm mirror height/width.

A further specification concerns the driving and resonant frequency of such a scanner. A high resonant frequency allows for a broad non-resonant driving-frequency range. This enables a higher LIDAR frame rate but also leads to greater mechanical robustness. Indeed, immunization to mechanical vibrations is a critical point for automotive-grade products, and resonant frequencies higher than 0.8 kHz are typically required for such applications [12].

Finally, we also selected the mirror design to operate in a quasi-static mode operation. While resonant mode scanners have the advantage of a larger deflection angle at given driving voltage as compared to quasi-static scanners, their operation is very sensitive to any environmental changes, such as temperature, increase for example, resulting in a shift of the resonant frequency. Without complex electronics, the induced mismatch between the resonance and drive frequencies leads to significant variability of the beam angle. By comparison, a biaxial quasi-static scanning mode is much less sensitive to environmental conditions. Moreover, this mode has the advantage of higher flexibility in terms of scanning frequencies and adaptive Field of view (FoV). Indeed, it offers the interesting capability to scan both the entire scene and also to focus on selected portions of the scene with, if necessary, a higher scanning frequency to increase the LIDAR resolution. Additionally, the control electronics are much simpler. The achievable scanning angles are, however, smaller compared to those achievable in the resonant mode.

In this work, as reported in Table 1, we designed the micro mirrors with a resonant frequency of about 2 kHz, and the targeted operational quasi-static fast-axis scanning frequencies was 600 Hz giving a 12 Hz frame rate. These values correspond to 10.000 points per frame which is consistent with the desired application.

### 2.2. Mirror Design and Bragg Reflector

Square 2D micro-mirrors supported by four suspended PZT actuator arms have previously been reported in [13,14]. Such symmetric configuration, shown in Figure 1, was selected among existing piezoelectric designs for this work, as it has the potential to meet the system specifications. In addition, this configuration is well suited to our application because the fast and slow axis each have a pair of dedicated actuators. This reduces the mutual impact of the two axes and simplifies the driving circuit. Concerning the component size, the reflective mirror is a 2 × 2 mm^2^-side square, and the MEMS size (mirror with PZT actuators and cantilevers) covers a surface of 8 × 8 mm^2^.

As previously discussed, the design was optimized to have resonant frequencies higher than 0.8 kHz. The three first resonance frequencies were calculated by modal analysis using COMSOL^®^ software. As shown in Figure 1, the first resonance mode is a pumping mode at 1511 Hz, in which the four-actuator beams vibrate in phase. The second and third resonance modes are rotational modes (similar to X- and Y-axis) at 2180 Hz, in which the two actuators of the same axis vibrate out of phase.

In order to fulfil the optical flatness criterion and to limit the dynamic deformation of the mirror as defined by [15] for a maximum drive frequency of 2 kHz and an optical angle of 5°, a mirror thickness of 20 µm was chosen.

Dielectric reflective coatings, known as Bragg coatings, have been used in this study to limit the incident power absorption and so to allow higher laser power. For a 1550 nm incident wavelength, the Bragg coating is composed of a repeated dielectric bilayer, made up of amorphous silicon (a-Si, 110 nm) and oxide (SiO_2_, 305 nm). This bilayer structure can be repeated significantly exceeding the reflectivity of metallic coatings at the operating wavelength. Bragg reflective coatings have, theoretically, no intrinsic absorption as the materials of the bilayer should be perfectly transparent at 1550 nm. In practice, a residual absorption exists largely due to interface defects, but it can be much lower than for a gold reflector.

### 2.3. Fabrication Process

To evaluate the Bragg reflector and its compatibility with the piezoelectric actuation technology, MEMS mirrors were fabricated with both conventional gold and dielectric multilayer reflectors as shown in Figure 2. The two types of reflectors were deposited on the same mirrors to compare their performances for otherwise identical designs and thicknesses.

As shown in Table 2, the fabrication of the 2D scanning mirror was performed on a (100) 8”-SOI substrate with a 20 µm-thick silicon layer, a 1 µm-thick buried oxide layer and a 700 µm-thick silicon handle wafer. (a) A 0.5 µm-thick SiO_2_ layer was first deposited by PECVD (Plasma-Enhanced Chemical Vapor Deposition) on the entire surface of the wafer, and then densified at 800 °C annealing under O_2_. The bottom TiO_2_ (20 nm)/Pt (100 nm) electrode was then applied by PVD and oxidation for TiO_2_ and by PVD (Physical Vapor Deposition) for Pt. The PZT piezoelectric film was then deposited using a sol-gel chemical solution deposition (CSD) method. The 0.535 µm thick PZT film is made up of 10 layers using a commercial PZT (52/48) solution provided by Mitsubishi Materials Corporation. Each layer of PZT is spun, dried at 130 °C and calcinated at 360 °C. This deposition is an automatic process. A first crystallization step is performed on the first deposited layer to promote the (100) orientation which is the best for optimal piezoelectric properties. Other crystallization steps are performed every three layers at 700 °C for 1 min under O_2_ in rapid thermal annealing (RTA) furnace. This PZT deposition is an automatic process. Next, the Pt (100 nm) top electrode was applied using PVD sputtering. Then (b), the top electrode, the PZT thin film and the bottom electrode were successively patterned using Reactive Ion Etching (RIE) via separate lithography steps.

For the fabrication of the gold reflector, a PECVD TEOS SiO2 layer (c) was deposited and patterned to prevent a short circuit between PZT electrodes. Then (d), the gold reflective layer and metallic contacts, a 700 nm monolayer, were deposited on the entire surface and patterned using wet etching then RIE (e).

For the fabrication of the Bragg reflector, an initial unsuccessful attempt at the bilayer deposition of the reflector was carried out using PECVD leading to a significant degradation of the PZT performance. Upon investigation, this was found to be the diffusion of hydrogen present during the PECVD deposition. The diffusion of hydrogen into the PZT and its electrodes resulted in a weakened interface and subsequent electrode delamination, as previously observed in [16]. To solve this problem, the Bragg reflector stack was deposited by PVD (Physical Vapor Deposition). At first, the 0.5 µm thick densified SiO2 PECVD layer has been selectively etched from the substrate at the desired mirror site (b). Then the Bragg bilayers of amorphous silicon (110 nm) and oxide (305 nm) were deposited n-times (c) depending on the desired reflectivity and absorption. An AMAT ENDURA magnetron sputtering chamber was used, loaded with a silicon target for the 110 nm amorphous silicon layers. The 305 nm silicon dioxide layers were deposited in the same chamber using reactive sputtering with a DC-pulsed power-generated plasma. Target oxidization hysteresis is managed by tunning oxygen flow and DC parameters, such as duty cycle and frequency. These bilayers were, then, selectively etched leaving the mirror structure. The mirror was etched down to the top of the first PVD oxide which prevented an electrical short circuit between the PZT electrodes (d). Metallic contacts were then deposited and patterned.

Both processes were completed by etching the SiO_2_/Si/SiO_2_ stack on the front side of the mirror using a dry etching process and finally a deep dry etch of the silicon on the backside of the mirror is performed to release the structure (e).

## 3. Results and Discussions

### 3.1. Piezoelectric Film Properties

Electrical and morphological characterizations of the PZT material were performed after fabrication. The deposited PZT film was found to be within specifications in terms of thickness (0.535 µm) and crystalline orientation (100).

Table 3 shows the capacitance voltage C-V curve and the $d_{31}$-V hysteresis loop. These curves are obtained at the end of the process with both gold and Bragg bilayer reflectors (2 bilayers). The C-V relation enables the extraction of the relative permittivity value *ε*r at 0 V, whereas the maximum piezoelectric coefficient *d*_31*max*_ is derived from the *d*_31_-V loop. The transverse piezoelectric coefficient $d_{31}$ represents the induced in-plane strain per unit voltage applied perpendicularly to the film plane [17].

The data showed nominal material behaviour for both the gold mirror (a) and the Bragg PVD process (b). The *ε*_r_ and *d*_31*max*_ coefficients were close for both technologies with values of 1370, 150 pm/V and 1390, 161 pm/V for the gold and Bragg PVD mirror, respectively. These results demonstrate that the process with the Bragg PVD mirror is fully compatible with the PZT process.

For comparison, the results obtained with the Bragg reflector using PECVD deposition are reported (c). They clearly show a drastic degradation of the performance of the PZT actuator on both C-V and $d_{31}$-V curves. *ε*_r_ and *d*_31*max*_ values are, respectively, equal to 1000 and 100 pm/V. These results were confirmed qualitatively by the non-operability of these mirrors.

### 3.2. PVD Bragg Reflector Results

Reflectivity (*R*) and transmission (*T*) were measured at a wavelength of 1550 nm for between one (n = 1) and four (n = 4) Bragg bilayers and for the gold reflector. These measurements were performed using a Cary 7000 spectrophotometer from Agilent set up with a 45° incident angle. R(%) and T(%) represent the percentages of incident power (a few mW) reflected by the mirror and transmitted through the mirror, respectively. The absorption A, which is the percentage of incident power absorbed in the mirror, is deduced from R and T using:$$A\left(\%\right)=100-R\left(\%\right)-T\left(\%\right)$$

Both orthogonal linear polarizations of the incident field were applied: p (incident electric field parallel to the plane of incidence) and s-polarization (incident electric field perpendicular to the plane of incidence). Figure 3 reports the values of R and A for the gold reflector and the n-Bragg bilayers (with n = 1 to 4). In each case, values reported for random polarization were calculated as the average of values measured with p- and s- polarization. Several results can be highlighted: firstly, the absorption value for gold random polarization is equal to 1.2%, i.e., 1.2% of the incident power is absorbed into the mirror and converted into heat. Secondly, Bragg mirror reflectivity is seen to approach 100% for higher numbers of Bragg bilayers. Concerning absorption, Bragg bilayers enable between a 1.5× (n = 4, p-polarization) to more than 24× (n = 4, s-polarization) decrease in absorption compared to the best metallic reflector (gold). The increase in absorption with the number of bilayers can be attributed to the defects present at the layer interfaces (discontinuity zone). These defects can act as absorption centers, and an increasing number of bilayers leads to more absorption centres. Improvements to the process could lead to the reduction of these defects and, therefore, lower absorption in the mirror.

For LIDAR application, random polarization of incident laser has been chosen to simplify the laser set up and to limit system cost. For the implementation of the micro mirror into the long-range LIDAR system, low absorption was chosen over a high reflectivity, i.e., the double bilayer configuration (n = 2) was selected.

It can be noted that if an s-polarised incident laser was to be used, the four Bragg bilayer (n = 4) would have been chosen because it provides very high reflectivity (99.91% vs Au: 99%) and a very low absorption (0.04% vs. Au: 0.98%).

### 3.3. Vibration Modes and Natural Frequencies

The resonance frequencies and vibration mode shapes characteristic of the 2D scanner were measured using a Vibrometer MSA 400 from POLYTEC mounted on a CASCADE prober from MICROTECH. The input signal is a dynamic driving voltage that can be defined as $V_{input}=V_{DC}+V_{AC}\sin\left(2\pi ft\right)$ where $\;V_{DC}$ represents a constant poling voltage, $V_{AC}$ the harmonic driving voltage magnitude at frequency f and the time t. The frequency (f) is swept from 10 Hz to approximately 2.5 kHz. Two driving configurations were successively applied. The first with all actuators driven with the same signal to excite the Z-pumping mode and the second with just two opposite PZT actuators driven with the same amplitude but with an opposite phase which enables the rotational modes to be excited.

For the Z-pumping mode, the vertical displacement (m/V) and the phase shift (°) measured at the centre of the mirror are reported versus the frequency in Figure 4. The phase shift (°) represents the difference between the reference-driving signal and the mirror response. Measurements were performed for $V_{AC}=V_{DC}=1.5\;V$. As shown, the Z-pumping mode resonance occurs at 1580 kHz.

The measurement of the rotational mode was not possible using the POLYTEC Vibrometer due to the angle limitation on the collection of the reflected signal. The measurements were performed using the projection of a Class 2 visible laser. The laser was deflected by the rotation of the mirror (on the X-axis) and projected on a screen. A camera recorded the projected position of the centre of the laser spot. The movement of the beam on the screen, due to mirror rotation, enables the optical angle to be deduced. The optical angle is defined as the total scanning angle of the reflected beam in X and Y-axis. This represents twice the total mechanical angle of the MEMS mirror. These measurements were performed as a function of increasing frequency with $V_{AC}=V_{DC}=2\;V$, and the uncertainty was measured to be ±0.2°.

As reported in Figure 4, the evolution of the optical angle (▲-black triangle) and phase shift (

-red dots) indicates a resonant frequency in the range [2450–2600] Hz. The precise value is limited due to coupling between X- and Y-axis natural modes whose natural frequencies are similar. Close to the natural frequency, cross talk disturbs the scanning and makes precise measurement difficult. For the following analysis, the resonant frequency was taken to be its lowest potential value, i.e., 2450 Hz. These two results are in good agreement with simulations.

The quality factor (Q factor) of the first pumping-mode was estimated to be ≈100, using the equation, $Q=\frac{\Delta f}{f_{r}}$, with Δf the peak width at the half maximum and $f_{r}$ as the resonant frequency.

As previously discussed, the resonant frequencies of the 2D scanner are above 1 kHz which is compatible with values typically required for MEMS mirrors for automotive LIDAR.

### 3.4. 2D Beam Scanning Results

Similar to that reported in [18], a sinusoidal scanning mode was tested, with a 600 Hz sinusoidal waveform for the fast axis and a 10 Hz ramp for the slow axis. In each case, the two opposing actuators have an opposite phase signal so as to cause the scanner to rotate around X and Y. In Figure 5, the 2D scanning action, projected on the screen, is represented with 50 points inside a 600 Hz period. The maximum PZT driving voltage used is 25 V.

Similar results are obtained for gold and Bragg bilayer coatings (n = 2). As shown, the X and Y-axis optical angles, with a 25 V applied voltage, are, respectively, equal to θx(°) = 4.5°(±0.2°) and θy(°) = 4.2°(±0.2°) for Bragg reflector (n = 2). For Gold reflector, values of θx(°) = 4.5°(±0.2°) and θy(°) = 4.3°(±0.2°) are obtained. The optical angle θx is measured at y = 0, and θy is measured at x = 0.

These results should be compared to previously reported values, such as those in [3] showing results obtained with 2D non-resonant scanning MEMS mirrors using electrostatic, electromagnetic, thermal and piezoelectric actuation. A figure of merit (FoM) can be used, which combines several key-parameters. $FoM=\theta_{e}\times d_{e}.\times f_{e}$ with $\theta_{e}$ the effective optical scanning angular field of view in radian, $d_{e}$ the effective dimension of the mirror in mm and $f_{e}$ the effective resonant frequency of the MEMS mirror in kHz. $d_{e}$ is defined as $d_{e}=\sqrt{\frac{4A}{\pi }}$ with A the surface of the mirror. $\theta_{e}$ and $f_{e}$ are defined as $\theta_{e}=\sqrt{\theta_{x}\theta_{y}}$ and $f_{e}=\sqrt{f_{x}f_{y}}$ with $\theta_{x}$, $\theta_{y}$ the maximum scanning angles in the X-axis and Y-axis, and $f_{x},\;f_{y}$ the resonant frequencies in the X-axis and Y-axis respectively. Using this FoM, our design variant 1 has *D* = 2 mm so a value of $d_{e}=2.25\;\mathrm{mm}$, $f_{e}=2.45\;\mathrm{kHz}$ (the design is symmetric, so the two resonant frequencies $f_{x}$ and $f_{y}$ are similar) and $\theta_{e}=0.076\;\mathrm{rad}$. For this 2D scanner FoM = 0.42. Only this first mirror design (Design variant 1) has been integrated and tested in the LIDAR system as reported below in paragraph 3.6. In parallel, optimisations based on the same design but with different hinge widths and lengths were implemented, fabricated and characterized, both with Bragg (n = 2) and Gold reflector (i.e., Design variant 2 and 3). Additionally, smaller mirror diameter were tested to measure the effective angle *θe* and effective resonant frequency *fe*, as shown in Figure 6. In the first case, design variant 4 with D = 1 mm, and in the second case, design variant 5 with *D* = 0.5 mm. The mirror characterisation was performed at a scanning frequency (fast axis) of 200 Hz and at a higher scanning frequency (close to 1000 Hz, 4000 Hz and 16,000 Hz).

Table 4 and Figure 7 show that the results we obtained are at the state-of-the-art with FoM-values close to 0.6–0.7 in the best case, comparable to thermal, electrostatic and electromagnetic scanners. In our best case, using design variant 3, an optical angle close to 8° was obtained with a 2 mm mirror (*D*). Additionally, we obtained the same level of optical angles both for gold and Bragg reflector technologies.

### 3.5. LIDAR System Integration and Results

A dedicated scanner packaging was developed to be fully compatible with an existing LIDAR system. The packaged MEMS mirror (Design variant 1) was integrated, as shown in Figure 8, using a wide-angle optical lens to expand the scanning angle up to 20°. LIDAR images have been obtained at 1064nm for an intermediate test before a final demonstration which will be performed at 1550 nm. The beam divergence at the laser output is lower than 0.4° and the laser had a pulse duration of 2 ns, a pulse repetition rate (PRR) of 120 kHz, and an average power that was below the class 1 eye-safety limit. The LIDAR system measures the round-trip delay from the light source to the receiver via the target to deduce the target distance. In Figure 9, from left to right, a scene is shown (a), the colour image (b) corresponds to the reflected light intensity at the laser wavelength, i.e., 1064 nm, and in (c), each point is coloured according to its distance. Finally, a 3D representation of the environment known as a point cloud (sets of points with 3D coordinates) is presented in Figure 10. These results clearly demonstrate the scanning stability and operability of our MEMS mirror.

## 4. Conclusions

In this paper, we report the development of a 2D non-resonant scanner designed for a LIDAR application at 1550 nm. Using VLSI technology, a 2D scanner was developed with PZT piezoelectric actuators and PVD Bragg reflective coatings. The study demonstrated that mirrors with a PVD Bragg reflector have the same level of performance as those using a gold reflector with the advantage that Bragg reflectors can manage higher optical powers. This is possible because the reflector absorption is reduced by a factor of up to 24 in the best case compared to a gold reflector. The use of a PZT actuator enabled a total optical angle close to 8° for a maximum driving voltage of 20 V. These results were obtained using a 2 × 2 mm^2^ square mirror. The performance of these devices is at least equivalent with respect to previously published demonstrations as well as being compatible with the criteria of mechanical vibration immunization which is essential for automotive applications. Finally, the MEMS mirrors have been packaged and integrated successfully in a functioning LIDAR system.

## Figures and Tables

**Figure 1 micromachines-14-01019-f001:**
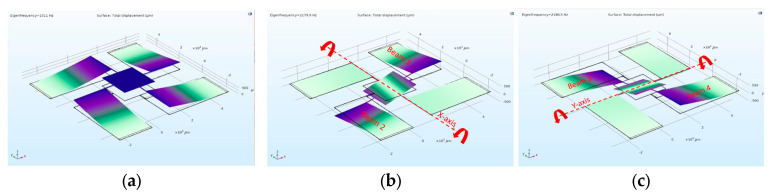
Modal analysis of the mirror: (**a**) First resonance mode (1.51 kHz-pumping mode); second (**b**) and third (**c**) resonance modes (2.179 and 2.18 kHz–X/Y-rotational modes)–Design variant 1.

**Figure 2 micromachines-14-01019-f002:**
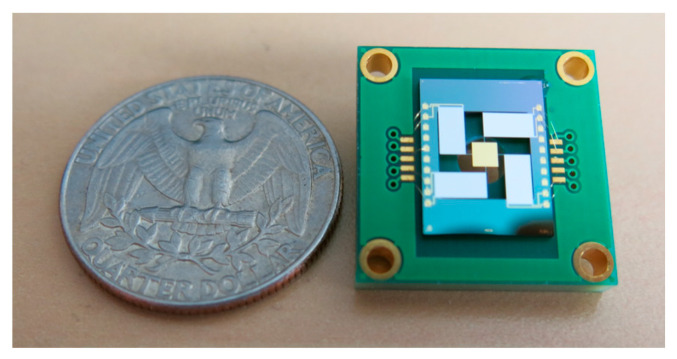
MEMS mirror top view.

**Figure 3 micromachines-14-01019-f003:**
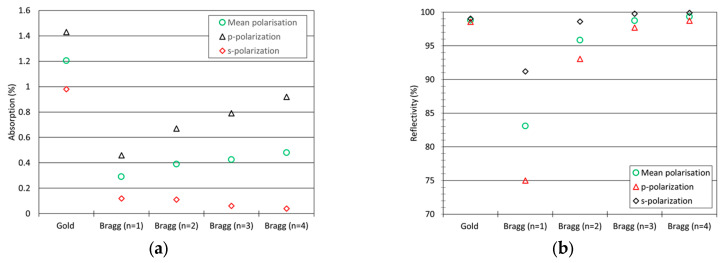
Evolution of absorption (%) (**a**) and reflectivity (%) (**b**) of PVD Bragg bilayer mirror vs the number of Bragg bilayer–comparison with gold reflective layer.

**Figure 4 micromachines-14-01019-f004:**
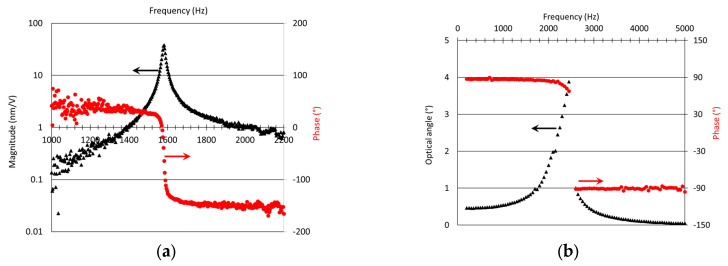
Magnitude/optical angle (▲) and Phase shift (●) for the first (Vpp = 3 V) (**a**) and second and third (Vpp = 4 V) (**b**) resonant modes–design variant 1.

**Figure 5 micromachines-14-01019-f005:**
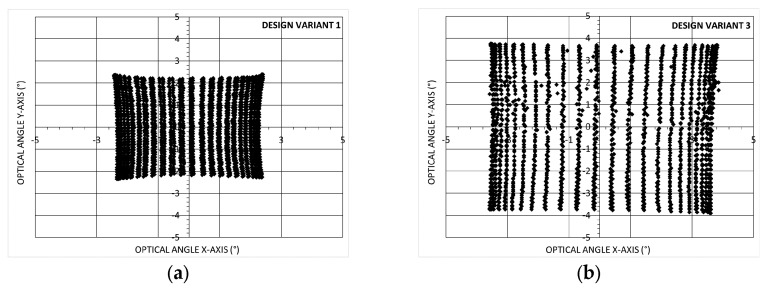
2D scanning representation of MEMS mirror with Bragg (n = 2) reflector-50 point per fast axis period–(**a**) Design variant 1:25 V voltage-600Hz fast/horizontal axis and 10 Hz ramp slow/vertical axis–(**b**) Design variant 3:20 V Voltage–200 Hz fast/Horizontal axis and 4 Hz ramp slow/vertical axis.

**Figure 6 micromachines-14-01019-f006:**
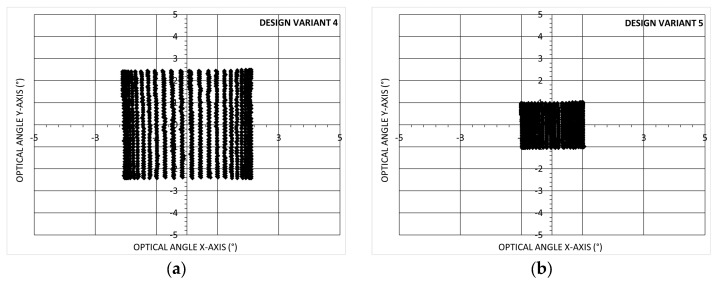
2D scanning representation of MEMS mirror with Bragg (n = 2) reflector-50 point per fast axis period–20 V voltage-200 Hz fast/horizontal axis and 4 Hz ramp slow/vertical axis-(**a**) Design variant 4; (**b**) Design variant 5.

**Figure 7 micromachines-14-01019-f007:**
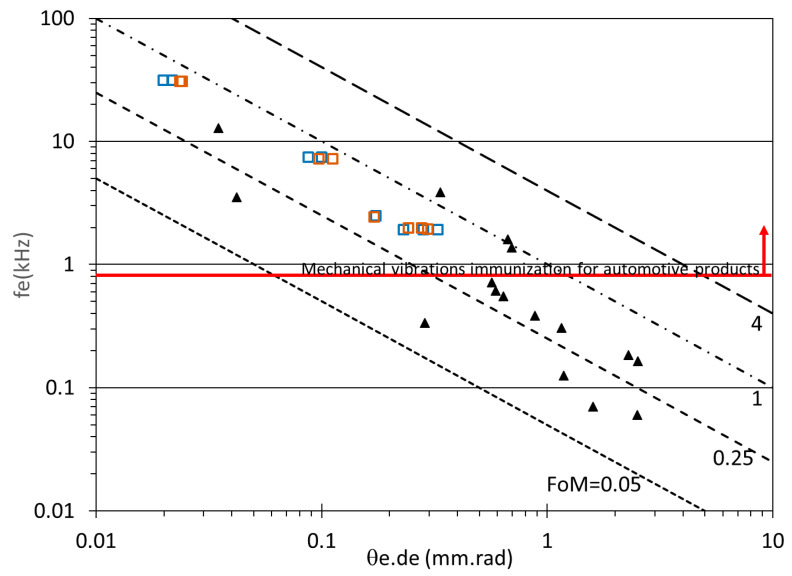
FoM values previously reported in [3] (▲) and this work (

 Bragg (n = 2) and 

 Gold reflector).

**Figure 8 micromachines-14-01019-f008:**
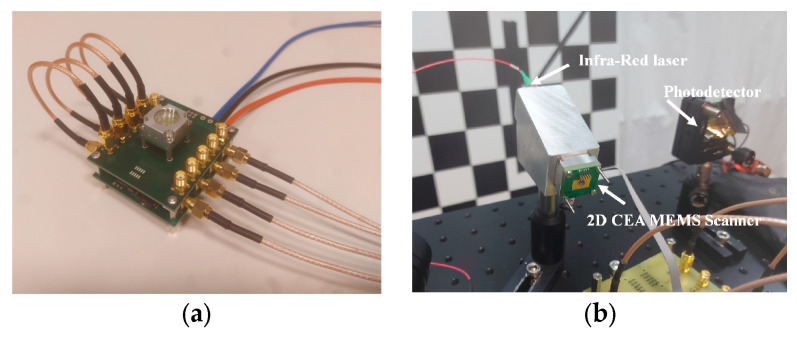
(**a**) Mirror packaging and (**b**) integration of the MEMS scanner inside Biaxial LIDAR system (Right).

**Figure 9 micromachines-14-01019-f009:**
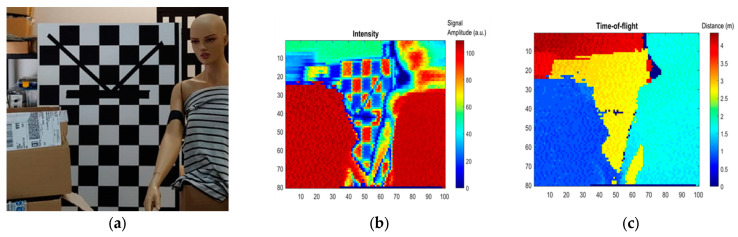
(**a**) View of the scene, (**b**) reflected laser intensity at 1064 nm and (**c**) depth value image.

**Figure 10 micromachines-14-01019-f010:**
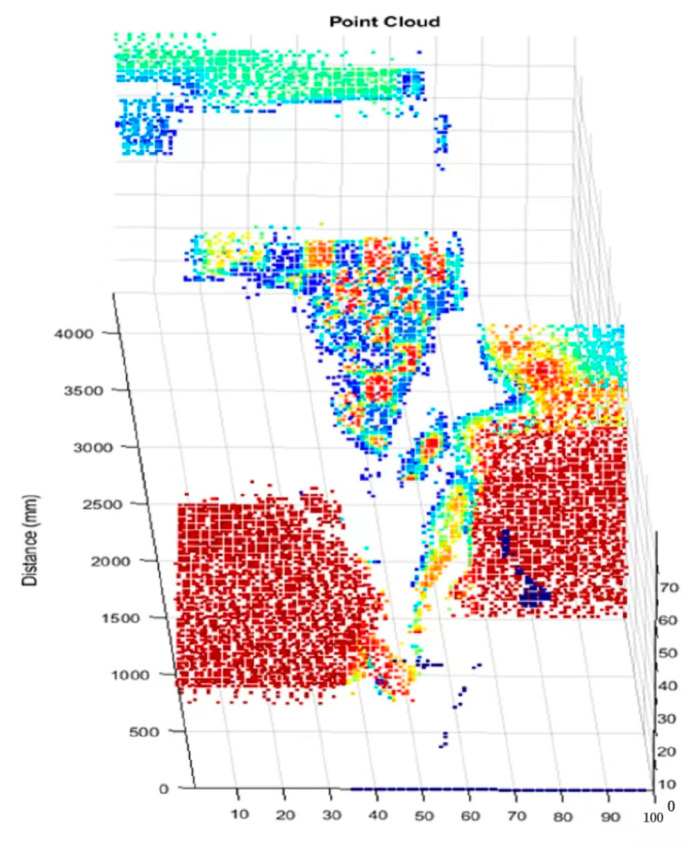
3D representation of the scene, Point Clouds.

**Table 1 micromachines-14-01019-t001:** EMS mirror specifications.

Mirror Diameter	Fast Axis (Quasi-Static)	Frame Rate (Quasi-Static)	Resonant Frequency
2 mm	600 Hz	Ramp 12 Hz	2 kHz

**Table 2 micromachines-14-01019-t002:** Description of the cross-sectional process flow for Gold and Bragg process flow.

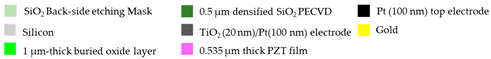
	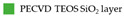	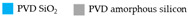
Step	Cross-Section Gold Process	Cross-Section Bragg Process
(a)		
(b)	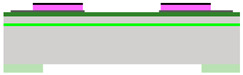	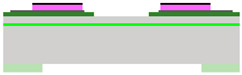
(c)	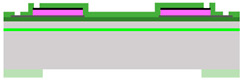	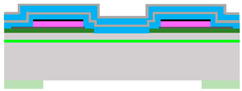
(d)	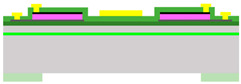	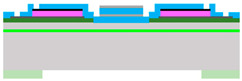
(e)	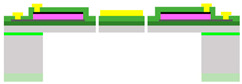	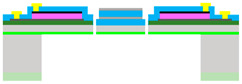

**Table 3 micromachines-14-01019-t003:** Comparison of PZT properties ($\varepsilon_{r}$ and $d_{31,max}$) for gold mirrors (a), PVD Bragg (b) and PECVD Bragg (c).

Reflector	C-V Curve	$d_{31}$-V Curve
(a) Gold process	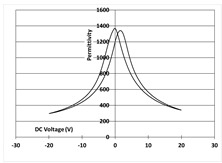	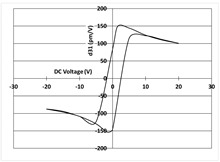
(b) Bragg PVD (n = 2) process	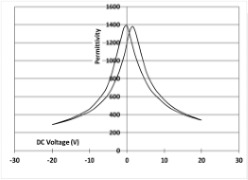	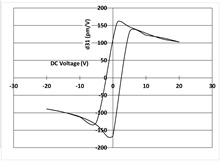
(c) Bragg PECVD (n = 1) process	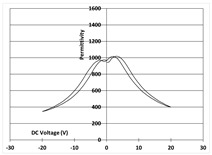	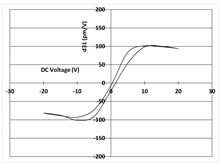

**Table 4 micromachines-14-01019-t004:** Design variant reported in this work with mirror size *D* (mm), MEMS length (mm) (mirror with PZT cantilevers and hinges), driving voltage and scanning frequency (Fast/horizontal frequency) (Hz) used for characterizations and measured values of *θ_e_*, *f_e_* and *FoM*.

Design Variant	Reflector	Mirror Size (*D*)	MEMS Side-Size	Driving Voltage	Scanning Freq. (Fast)	Non-Resonant Angle *θ_e_*	*f_e_*	*FoM*
	Au/Bragg(n)	(mm)	(mm)	(V)	(Hz)	(°)	(kHz)	
1	Au	2	≈8	25	600	4.4	2.45	0.42
1	Bragg (2)	2	≈8	25	600	4.3	2.5	0.43
2	Au	2	≈8	20	200	6.1	2	0.48
2	Bragg (2)	2	≈8	20	200	5.8	1.92	0.44
3	Au	2	≈8	20	200	7	1.95	0.54
3	Bragg (2)	2	≈8	20	200	7.1	1.92	0.54
3	Au	2	≈8	20	880	7.5	1.95	0.57
3	Bragg (2)	2	≈8	20	1000	8.2	1.92	0.62
4	Au	1	≈4.1	20	200	4.9	7.25	0.7
4	Bragg (2)	1	≈4.1	20	200	4.4	7.5	0.65
4	Au	1	≈4.1	20	4000	5.65	7.25	0.8
4	Bragg (2)	1	≈4.1	20	4000	5.06	7.5	0.74
5	Au	0.5	≈2.3	20	200	2.4	31	0.74
5	Bragg (2)	0.5	≈2.3	20	200	2	31.5	0.62
5	Au	0.5	≈2.3	20	16,000	2.4	31	0.72
5	Bragg (2)	0.5	≈2.3	20	16,000	2.2	31.5	0.68

## Data Availability

Not applicable.

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
