# Peer review of "Biaxial Piezoelectric MEMS Mirrors with Low Absorption Coating for 1550 nm Long-Range LIDAR"

_micromachines, 2023, doi:10.3390/mi14051019_

Round 1

Reviewer 1 Report

This paper did a simple improvement for the PZT MEMS mirror by changing the reflector into Bragg reflector. Considering the shown works and submitted paper, I cannot give the work a positive recommendation. 
1.The novelty and motivation are not well shown. I cannot get the real value of this work from the Introduction and other sections. The fabrication, the new reflector or new mirror? The introduction should be greatly improved to show the state-of-art for your work and the existing problems to show the value of your work.
2.The paper should be checked before its submission. Obvious errors happen in the manuscript.
3.The paper says the Bragg reflector can decrease the absorptivity from 0.98% to about 0.2%. I am wondering the real influence of this improvement on the imaging quality of LIDAR by experiment. Moreover, the tested n=2 device seems to have a higher absorptivity than Au. Then, what’s the real target of this paper?
4.Then a flow diagram for the fabrication of mirrors with Au and Bragg reflectors should be clearly given, especially for the Bragg multi-layers to prove the words “being compatible with CMOS technology”.
5. How to get thermal SiO2 by PECVD? Then, how to get TiO2/Pt electrode by PVD?

6. The caption of Fig.5 cannot indicate the results for Magnitude and Phase shift.

7.The title give a wavelenght of 1550nm, but the test is conducted at 1064nm. Why not use the laser in title? 

Author Response

General considerations

The publication has been refocused on the development of a Bragg process compatible with the piezoelectric motor, which forms the core of the paper.

To this end, the Bragg PVD process has been detailed and compared to the Au process. Similarly, the tests performed with the Bragg PECVD process have been reported in terms of the PZT performance in order to clearly highlight the problem and the solutions we have provided.

In addition, the publication has been reviewed and edited by a colleague who is a native English speaker.

  1. The introduction has been completely rewritten to focus on the problem of Bragg development in PVD and the results obtained in terms of absorption for the reflective layer. The introduction also highlights the limitations and problems of using a metal reflector and the gain of a low absorption reflector for a LIDAR application.
  2. The paper has been completely rechecked.
  3. Figure 3.a shows that under our test conditions (average polarisation), the absorption decreases from 1.2% (Au) to 0.4% (n=2, Bragg PVD) which represents an absorption decrease of a factor of 3. Thus, this reflective layer will limit the heating of the mirror and therefore its thermal deformation (which directly impacts the reflected beam). It will also allow to keep the performance of the PZT (which can degrade at high temperatures). These low-absorption layers allow the incident power to be increased above 2W for very long range LIDARs or other high power applications.
  4. Table 2 provides a clear comparison of the Au and Bragg technological processes. Regarding the compatibility with CMOS, the sentence has been deleted as it is outside the scope of the paper. Indeed, the advantage of Bragg PVD technology is that it does not use non CMOS compatible materials such as Au. However, in the case of sol-gel deposition with annealing at 700°C, the thermal budget is a limitation. Solutions exist, notably with a reported PZT technology. But this is beyond the scope of this paper.
  5. Figure 5 has been corrected.
  6. The tests could not be carried out at 1550 nm because the 1550 nm detector, supplied by a project partner, was not functional… The tests therefore had to be carried out at 1064 nm and enabled the stability of the 2D scan to be verified.

Reviewer 2 Report

System specifications, referenced throughout the text, for the reader’s benefit, are better to gather in a single table.

Fix the Figure 1 Caption.

Line #

Comment

111-112

Eliminate the figure

121-122

Eliminate the figure

133-134

Provide some details related to the stress-compensated technology used, and what the procedure consists of.

138

Get rid of the space before the line

143

Provide higher-resolution images

167

Correct the title number

172

The technological process depicts the Bragg/reflector case, it is recommended to add the gold to the Figure 2b legend (“Amorphous Si/Gold”) for the sake of full clarity, given the text nominates both cases. Please provide more distinguishable colors for bottom electrode and silicon.

In addition, please specify explicitly, either in the text or in the figure legend, which layers serve as stress-compensating (all three that support the reflective layer or just the adjacent one?), given the fact that among the core contributions of the work is the technology itself. The detailed research regarding the contribution degree of the layers in question may also be interesting for future work.

234

Switch the (a) and (b) assignments in Figure 3 for reflectivity and absorption in correspondence with the caption.

294

Eliminate the figure

298

Being optical angle and phase shift represented in different colours on the Figure 5, provide the same colours in the caption as well.

308

Put the parenthetical word into a passive mode (“As shown...”).

321

Correct the multiplication sign in the FoM, in the present form it seems to be a superscript of e.

326-327,331-332,334-335

Eliminate the spaces between the lines

412-414

Control the consistency of the reference provided

Author Response

General consideration:

The publication has been refocused on the development of a Bragg process compatible with the piezoelectric motor, which forms the core of the paper.

To this end, the Bragg PVD process has been detailed and compared to the Au process. Similarly, the tests performed with the Bragg PECVD process have been reported in terms of the PZT performance in order to clearly highlight the problem and the solutions we have provided.

In addition, the publication has been reviewed and edited by a colleague who is a native English speaker.

Table 1 has been added to summarise all the specifications expected of the scanner

Line 111 à The figures have been eliminated

Line 121 à The figures have been eliminated

Line 133 à The paragraph on stress compensation technology has been removed as it does not form the core of the publication and does not provide any interesting insights. The aim of the publication is to highlight the problems associated with a metal reflector and the challenges of implementing a process that is compatible with Bragg reflector and piezoelectric motor.

Line 138 à Done

Line 143 à Done

Line 167 à Corrected

Line 172 à Table 2 provides a clear comparison of the Au and Bragg technological processes. This makes it possible to clearly highlight the interest of the publication, ie the development of a Bragg process, compared to the reference process with gold.

Line 234 à Done

Line 294 à Done

Line 298 à Figure 4 has been modified as requested

Line 308 à Done

Line 321 à It has been corrected

Line 326-335 à Done

Line 412 à References have been all controlled

Reviewer 3 Report

A realization and the characterization of a biaxial PZT-based piezoelectric MEMS scanner incorporates a Bragg reflector is reported in this paper. This manuscript is more like a technical report than a research paper. My concerns are listed as following.

1, How many DoFs (Degree of Freedom) can be achieved for this proposed mirror, and what about the workspace/reflecting scope of the mirror? It is better to provide theoretical analysis.

2, For Line 112 and 122, these two sentences are incomplete.

3, Explanations for presented figures are confusing since Figures titles in Page 3 and Page 8 are missing.

4, How fast scanning did this mirror can be achieved?

5, Motivations should be highlighted in Introduction Section.

6, References cited in introduction are so casual. Authors should pay more attention of format practice.

Author Response

General consideration:

The publication has been refocused on the development of a Bragg process compatible with the piezoelectric motor, which forms the core of the paper.

To this end, the Bragg PVD process has been detailed and compared to the Au process. Similarly, the tests performed with the Bragg PECVD process have been reported in terms of the PZT performance in order to clearly highlight the problem and the solutions we have provided.

In addition, the publication has been reviewed and edited by a colleague who is a native English speaker.

  1. The mirror can reach 3 DOF. These 3 DOFs are shown in Figure 1. In our scanning mode, we use the two rotational modes (b) and (c). The pumping mode, which we do not use, has been figured to indicate where its resonant frequency is. And to ensure that this frequency is greater than 1 kHz. Theoretical analysis have ever been reported in ref [5] and [6], and we did not extent on this point as the main development of the publication is compatibility between Bragg reflector and piezoelectric motor.
  2. Sentences of line 112 and 122 have been corrected
  3. Figures have been modified for clarity
  4. We are currently working on this point, in order to clearly define the maximum frequency at which we can use the mirror while keeping good control over it in quasi-static mode. The frequency we have reported is half the resonance frequency in each case (approximately). We are currently studying the 2D scanning figure and the evolution of crosstalk, with these mirrors and new mirrors, when the drive frequency is closer to the resonant frequency.
  5. The introduction has been completely rewritten to focus on the problem of Bragg development in PVD and the results obtained in terms of absorption for the reflective layer. The introduction also highlights the limitations and problems of using a metal reflector and the gain of a low absorption reflector for a LIDAR application.
  6. References have been all rechecked

Round 2

Reviewer 1 Report

The paper has been improved, and a minor problem is here.

The title and many other places show the key word ' 1550 nm'. However, the design  did not show special limitation from the number of 1550nm, and the experiments cannot verify the claimed feature of Bragg reflector under 1550nm light.

So, is it real necessary to underline the word  ' 1550 nm'?

Author Response

The Bragg reflector of the piezoelectric mirror was specifically developed for an incident wavelength of 1550 nm. This means that the material of each bilayer and especially its thickness were optimised, first by modelling and then experimentally, to maximise the reflectivity at this wavelength.

In addition, the absorption measurement was carried out at this wavelength and the resulting gain, a decrease of a factor of 24 compared to gold, was measured for this wavelength.

So I think it is important to mention the wavelength in the title of the paper. If we had developed a mirror at 905 nm (the other wavelength commonly used for LIDAR), the materials used and the thicknesses would have been different. And we would not have encountered the same problems.

Unfortunately, as I said earlier, the final tests (integrated in a LIDAR) could not be carried out because the detector (at 1550 nm), provided by a project partner, was not functional. However, this does not call into question the performance of the 2D mirror for this application.

Additionally, to clearly highlight the state-of-the-art I added 7 latest references in the field when revised

Reviewer 3 Report

This manuscript is more like a technical report, rather than a scientific research paper.  In addition, some writings cannot meet research writing requirements. For example, in page 1, line 33, "such as [1]", in page 3, line 112, "as reported by [4]" looks very strange. Moreover, this manuscript lacks novelty, issues reported in this paper were already solved by many previous studies. Thus, references in this manuscript are not new. It make no sense to publish this manuscript.

Author Response

The line 33 in page 1 and line 112 in page 3 have been corrected as requested.

The core and novelty of the paper is the development of a Bragg process compatible with the piezoelectric motor. The novelty is not the design of the mirror, which is known.

As I suggest in the paper, in order to achieve higher beam power, we have developed a multi-layer non-metallic Bragg reflector that limits the absorption of incident power and is fabricated using a monolithic silicon fabrication process.

As reported, the development of a Bragg reflector on a 2D MEMS scanner has already been published, but with a different actuation technology (electrostatic) and for a different wavelength (1064 nm), which requires others materials for the Bragg reflector than those used in the paper [ref. 9].

The use of an electrostatic actuator technology does not present the same technical challenges, in particular the compatibility of the Bragg materials deposition with the piezoelectric sol-gel deposition.

To the best of our knowledge, this paper show for the first time the fabrication, using VLSI technology, of a 2D piezo-scanner designed for a 1550 nm LIDAR system, using a low-absorption dielectric reflective coating compatible with sol-gel deposition of the piezoelectric motor.

To clearly highlight the state-of-the-art I added 7 latest references in the field when revised

Round 3

Reviewer 3 Report

This manuscript can be considered for publication in current form.